# Building a Flower: The Influence of Cell Wall Composition on Flower Development and Reproduction

**DOI:** 10.3390/genes12070978

**Published:** 2021-06-26

**Authors:** José Erik Cruz-Valderrama, Judith Jazmin Bernal-Gallardo, Humberto Herrera-Ubaldo, Stefan de Folter

**Affiliations:** Centro de Investigación y de Estudios Avanzados del Instituto Politécnico Nacional (CINVESTAV-IPN), Unidad de Genómica Avanzada (UGA-LANGEBIO), Irapuato CP 36824, Guanajuato, Mexico; erik_cruz_v@hotmail.com (J.E.C.-V.); judith.bernal@cinvestav.mx (J.J.B.-G.); humberto.herrera@cinvestav.mx (H.H.-U.)

**Keywords:** flower development, cell wall, remodeling enzymes, cellulose, hemicellulose, pectin

## Abstract

Floral patterning is a complex task. Various organs and tissues must be formed to fulfill reproductive functions. Flower development has been studied, mainly looking for master regulators. However, downstream changes such as the cell wall composition are relevant since they allow cells to divide, differentiate, and grow. In this review, we focus on the main components of the primary cell wall—cellulose, hemicellulose, and pectins—to describe how enzymes involved in the biosynthesis, modifications, and degradation of cell wall components are related to the formation of the floral organs. Additionally, internal and external stimuli participate in the genetic regulation that modulates the activity of cell wall remodeling proteins.

## 1. Introduction

The process of flower development involves the formation of very complex structures. Flowers generally arise from the inflorescence meristem, where cellular differentiation processes give rise to specialized structures involved in all the aspects of plant reproduction. Once the flower primordium is formed, specific changes can be observed in this area; cell growth, division, and differentiation make up tissues with different identities and functions [1]. A large part of the angiosperms shows a relatively conserved arrangement of the floral whorls (Figure 1). From the inside out, we can find the gynoecium, the female structure of the flower which houses the ovules, surrounded by the androecium, the male part of the flower, represented by the stamens and, where pollen grains are produced. The corolla, made up of vegetative structures that present a wide range of colors, helping to attract specific pollinators, and the most outside whorl is the calyx, where the sepals protect the floral structures formed towards the center of the flower [2,3]. The formation of each whorl involves specific changes at the cellular level where cell division, expansion, and cell death intervene finely to generate the female and male gametes that will eventually come together in the fertilization process, giving origin to the seeds that ensure the next generation of plants (Figure 1). Flower development has been finely described for plant species such as Arabidopsis, Petunia, and Antirrhinum. For these species, several genes (most of them transcription factors) are known to guide the differentiation of the floral whorls [1,4]. Knowledge from these model species suggests that at the molecular level, there is a core mechanism that is conserved across the great variety of flowering plants [5].

The involvement of cell wall-related genes has been described during plant development. The detection of different cell wall components in different plant organs, including roots, leaves, stems, and flowers has been reported [6]. In addition, work has been performed analyzing cell wall composition and dynamics during carpel medial domain development [7]. The plant cell wall is a dynamic structure composed mainly of three types of polysaccharides: cellulose, hemicelluloses, and pectins. These polymers interact and assemble to constitute a network with structural proteins that are inserted in a gel-like matrix through different types of chemical bonds and physical arrangements [8]. This architecture allows plant cells to maintain a defined shape, confer mechanical support, and keep intercellular communication in the different organs. The composition and chemical structure of these polysaccharides may change during plant development in response to environmental and endogenous signals [9]. The composition of the cell wall varies depending on the species, the cell type, and even the subcellular domain within the cell wall of one single cell [9]. This variation in the content suggests that the plant cell wall is highly dynamic and can be modulated in quite specific ways. The cell wall plays an essential role in the morphogenesis of plants. As mentioned above, gene regulatory networks underlying the differentiation of the floral meristem and floral patterning control the expression of genes involved in the synthesis and modulation of cell wall structure, leading to dynamics in the cell wall structure that is necessary for proper flower development.

In *Arabidopsis thaliana* L., research has been conducted on genes involved in the formation, composition, and maintenance of the cell wall in the shoot apical meristem (SAM) [10]. The characterization of the *CELLULOSE SYNTHASE-LIKE D2*, 3 and 5 [11], the *CELLULOSE SYNTHASE 3*, *CELLULOSE SYNTHASE 1*, and *CELLULOSE SYNTHASE 6* genes lead to the discovery of their involvement in the biosynthesis of cellulose [12]. Furthermore, the *XYLOSYLTRANSFERASE 1*, *XYLOSYLTRANSFERASE 2*, and *α-XYLOSIDASE 1* genes were identified to play a role in the modifications of xyloglucans in the cell wall [10,13]. The absence of these enzymes leads to changes in the size, microtubule arrangement, phyllotaxis, geometry, and development of the SAM. On the other hand, overexpression of genes that encode enzymes such as Pectin Methyl Esterase 5 (PME5), involved in pectin modification, causes the formation of ectopic primordia, while the excess of Pectin Methyl Esterase Inhibitor 3 (PMEI3) inhibits organ formation [14]. These investigations highlight the importance of these genes in SAM development and the subsequent formation of the aerial organs of the plant.

In this review, we summarized recent and classical studies showing the influence of primary cell wall components on flower formation. We focused on the major components: cellulose, hemicellulose, pectins, and cell wall proteins. Most of the information presented here comes from Arabidopsis, though we also included some works on tobacco, rice, tomato, carnation, and other flowering species.

## 2. Cellulose

One of the main components of the cell wall is cellulose. Cellulose comprises glucose molecules linked by β-(1-4) bonds synthesized by cellulose synthase complexes located in the plasma membrane. The deposition of the cellulose microfibrils influences the direction of the cell wall growth and, for instance, is essential for anisotropic growth. In this process, cellulose synthase complex conformed by cellulose synthases and other accessory proteins may be orientated by cytoskeletal control [15].

It is interesting how a lack of cellulose synthesis can affect the development of floral tissues. An example in *Arabidopsis thaliana* is the mutation of the *RADIAL SWELLING* (*RSW1*) gene, also known as *CELLULOSE SYNTHASE 1* (*CESA1*) gene, which is involved in cellulose synthesis and when not functional causes defects related to reproductive development. These defects are the result of stigmas that protrude beyond the sepals and shortened petals. Although the anthers have normal dehiscence and pollen development, self-pollination is reduced because the stigma is well above the anthers as they have short filaments and show a wrinkled surface [16]. Interestingly, work showed that the null mutants of *CESA1* and *CESA3* genes are pollen defective [17]. These phenotypic alterations suggest an impact in cell growth with a lower amount of cellulose synthesized and a lower amount of generated biomass, resulting in smaller calyx and corolla whorls. Similarly, the reduced fertility shown in *CESA3* antisense plants is associated with shorter filament cells [18]. On the other hand, the characterization of the Arabidopsis Cellulose Synthase-Like D genes *CSLD1* and *CSLD4* showed their importance for pollen tube growth. Both *CSLD1* and *CSLD4* are highly expressed in mature pollen grains and pollen tubes. Mutations in *CSLD1* and *CSLD4* caused a significant reduction in cellulose deposition in the pollen tube wall and disorganization of the pollen tube wall layers. In *csld1* and *csld4* single mutants and the *csld1 csld4* double mutant, the pollen tubes exhibited abnormal growth both in vitro and *in vivo*, affecting the pollination capacity [19]. The importance of cellulose synthesis for pollen tube growth is also observed in the Arabidopsis mutant of the gene *AtCSLA7*, a member of a *CELLULOSE SYNTHASE-LIKE* (*CSL*) subfamily [20]. 

We can conclude that the amount of cellulose that is synthesized has a significant effect on flower function. Multiple factors could be indirectly involved in cellulose synthesis since its anabolism depends on the plant nutritional status. For example, plants with enhanced sucrose metabolism showed increased plant biomass and abnormal morphological flower phenotypes, suggesting that the physiological processes related to the nutritional status can modulate cell wall composition [21].

It is worth mentioning that other studies not explicitly related to flower development, showed that the orientation of the cellulose fibers within the cell wall is determinant for proper plant development. In one of these experiments, a compound called cobtorin was used to alter cellulose fiber deposition within the cell wall, which resulted in amorphous and swollen cells in roots and hypocotyls [22]. It would be interesting to know if this type of alteration occurs in floral tissues when disrupting cellulose synthesis.

The phenotypes mentioned above suggest that in addition to reducing the size of the floral organs, a consequence of decreased cellulose synthesis is abnormal pollen tube elongation, which ultimately affects fertilization and subsequent seed formation. Given the importance of cellulose as the main component in cell wall biosynthesis, it is interesting to analyze genes involved in cell wall biosynthesis that are expressed in floral tissues in Arabidopsis. Most of them have not yet been functionally characterized but many are expressed in floral tissues [23], as shown in Figure 2. 

## 3. Hemicelluloses

Another major component of the plant cell wall are hemicelluloses, which are constituted by a diverse group of sugars such as xylose, mannose, arabinose, galactose, or fucose. This group constitutes roughly one-third of the cell wall biomass and encompasses mannans, xyloglucans, xylans, and mixed-linkage glucans. The structure of these polysaccharides, particularly their substitutions, varies depending on the tissue and the plant species [24,25]. The dominant backbone linkage is represented by the β-(1-4) glycosidic bond, although β-(1-3,1-4) glucans are also considered hemicelluloses. These polysaccharides are synthesized from activated nucleotide sugars through different families of synthases and glycosyltransferases [24,25].

The structure and complexity of hemicelluloses are crucial factors affecting their activity and function. Synthesis (formation) and hydrolytic (degradation) reactions contribute to the overall distribution of a polymer size. Studies from the early 1990s showed the redistribution in polymer size of hemicelluloses and endoxylanase activity in a partially purified enzyme preparation of *Dendrobium crumenatum* petals [26], suggesting that degradation of hemicelluloses is also crucial in processes such as petal senescence.

Based on the work performed in Arabidopsis by Becnel et al. in 2006 [27], it is important to highlight the presence of transcripts of different xyloglucan endotransglucosylase/hydrolases (XTH) encoding genes was observed in the flower. The observed transcript distributions suggest the participation of each member in different floral whorls; for instance, *XTH1*, *XTH30*, and *XTH33* showed expression in anthers at different developmental stages. The *XTH29* gene showed expression in pollen. *XTH5* expression was detected in the distal region of the stamen filament, possibly related to elongation. On the other hand, *XTH9* showed the highest expression in the youngest flowers and loss of generalized expression as the flower matured, specifically in the terminal portions of the carpel and throughout the stamen filaments. *XTH15* and *XTH28* also showed complex expression patterns in the developing floral bud, with prominent expression in the carpel. On the other hand, *XTH28* was detected only in sepals [27]. The expression of the *XTH* genes seems to be tightly regulated at the transcriptional level, resulting in precise expression patterns.

These expression patterns of *XTH’s* suggest that they are involved in floral organ development; however, future functional work on these genes will be necessary to understand their involvement in floral development.

Studies from other species suggest the importance of hemicellulose modifications during the organ abscission processes. During floral abscission in tomato (*Solanum lycopersicum*), an increase in the presence of arabinans and xyloglucans in the pedicels was observed. In addition to this, a peak in the *XTH* presence in the abscission zone was observed one day after anthesis, which decreased over time. These results suggested that XTH enzymes may play an essential role in the floral abscission process [28].

During flower opening in carnation (*Dianthus caryophyllus* L.), the expression of two genes encoding for *DcXTH2* and *DcXTH3* was detected in large quantities in petals compared with other tissues. The function of XTH in growing petal tissues was verified by an in situ staining of xyloglucan endotransglucosylase (XET) activity, which was detected at all stages of flower opening in all parts of growing petals. However, the activity tended to be higher in the base and boundary regions [29]. Additionally, the gene *MANY SENESCENCE-RELATED 12* (*SR12*), putatively encoding a β-galactosidase, has been isolated from carnation petals and related as a marker of senescence-associated with ethylene [30]. The xyloglucan depolymerizing, β-glucosidase, and β-galactosidase activities were detected in protein extracts from senescing flowers of carnation; the enzymatic activity levels were consistent with the changes in the hemicellulose fraction of the cell wall found in the work performed by de Vetten et al., 1991 [26].

Studies related to the dynamics of the formation of hemicellulose have been linked to petal formation. In Sandersonia (*Sandersonia aurantiaca*), three genes encoding putative β-galactosidases (*SaGAL1*, *SaGAL2*, and *SaGAL3*) have been isolated from petal tissue. Expression of all three genes is observed in petals at the onset of flower wilting [31]. In Arabidopsis, putative α-L-arabinofuranosidase genes were detected in flowers, *α-L-ARABINOFURANOSIDASE 1* (*AtASD1*) detected in vascular tissues of mature and senescent petals, and *α-L-ARABINOFURANOSIDASE 2* (*AtASD2*) localized throughout the petal in fully mature flowers [32]. Likewise, in Arabidopsis, the characterization of the α-xylosidase gene *XYL1* showing expression in sepals, style, stamens, and the gynophore. The mutant of this gene causes altered growth of sepals [33]. Results from both Sandersonia and Arabidopsis for these genes suggest that they may be involved in different events of flower development.

In summary, one of the processes where hemicelluloses could be intervening is cell separation, which generally occurs during the maturation and senescence of the calyx and corolla, and flower abscission. However, more functional studies are required to demonstrate this activity forcefully.

## 4. Pectins

Flowering plant species generally have greater complexity in the conformation and regulation of cell wall components and have more actors than non-flowering plants. Evolutionarily, evidence suggests this idea, based on the presence of a greater variety of molecules that make up the cell wall and proteins with enzymatic activity that regulate the biochemical structure of its components in flowering plants [34,35]. 

Pectins are another type of polysaccharide present in the primary cell wall. Homogalacturonans (HGs) are the most representative group of pectins. They are made up of galacturonic acid monomers linked in α 1-4 bonds [36]. Rhamnogalacturonans I (RGIs) are a disaccharide skeleton made up of galacturonic acid and rhamnose. In contrast, Rhamnogalacturonans II (RGIIs) conserve the HG skeleton, but in their branches, they present a great diversity of polysaccharides that include rhamnose [36]. Pectin synthesis of all groups requires several enzymes that transfer sugars to the structure. Pectins are synthesized in the Golgi apparatus to be methyl esterified at their C-6 carboxyl group by pectin methyltransferases (PMT). The addition of side chains to the carbon skeleton is carried out in the trans region of the Golgi apparatus, generating Rhamnogalacturonans I and II and xylogalacturonans [36]. Once localized in the cell wall, the different types of pectins can be demethyl esterified by pectin methyl esterases (PME) [37]. These biochemical changes in the pectins can lead to the degradation of these pectins by other enzymes like polygalacturonases or pectate lyases [37]. Another chemical change that pectins can undergo are the O-acetylatation at the C-2 or C-3 position. The resulting acetylesters change dynamically during the growth and development of plants [38].

It is interesting how the lack of pectin esterification can affect gynoecium development. For instance, a rice mutant of a putative pectin methyltransferase (PMT) encoded by the *OsPMT16* gene showed an alteration in flower formation; the pistils were slightly shorter with delayed growth of the stigma and shorter stamens, which led to significantly reduced fertility [39]. These results indicate that pectin modification contributes not only to pistil development but also to reproductive capacity in rice. Regarding the removal of methyl groups from pectins, it has been reported that pectin methyl esterases (PME) play a role in pollen grain formation; for instance, the Arabidopsis mutant of the *QUARTET1* gene is affected in cell wall degradation, and the microspores remain grouped during microsporogenesis [40]. Another example in Arabidopsis is the gene *VANGUARD1* (*VGD1*) that encodes another pectin methylesterase protein. The *vgd1* mutant is affected in the growth of pollen tubes in the style and transmitting tract in the gynoecium. Genetic studies indicated that the *vgd1* mutant did not show visible effects in the rest of the flower [41]. Interestingly, when *PME5* is overexpressed, it can partially recover the phenotype of Arabidopsis plants where *ETTIN* (*ETT/ARF3*) is mutated, *ettin* is a mutant where the valves are affected in their size along the carpels in the gynoecium [42].

In another work performed to characterize another pectin methylesterase encoded by *AtPPME1*, it was shown that the *atppme1* mutant had alterations in pollen tube growth, length, and morphology. However, it had no apparent effect on pollen grain morphology and cell wall patterns [43]. Similarly, in *Nicotiana tabacum*, silencing of the gene *NtPPME1*, encoding the main tobacco PME isoform, showed a slight but significant decrease in in vivo pollen tube growth [44]. These results are consistent with the works in Arabidopsis. 

The enzymatic activity of PMEs is regulated by the inhibition of Pectin Methyl Esterase Inhibitors (PMEI). These PMEIs act by regulating the demethyl esterification of homogalacturonans affecting cell wall mechanics and consequently plant development. Some PMEIs show differential expression in inflorescences and floral tissues of Arabidopsis, suggesting that they could play specific roles in flower development [34,45]. More specifically, it was found that *AtPPME1* and *AtPMEI2* are both expressed in Arabidopsis pollen; furthermore, the proteins physically interact, and that AtPMEI2 inactivates AtPPME1, at least under in vitro conditions. Additionally, in the same work, research performed with transient expression of the *AtPMEI2* in tobacco pollen tubes showed increased length of pollen tubes compared to wild type, suggesting its role in cell expansion [46]. In *Brassica oleracea* L. var. *italics*, the pectin methyl esterase inhibitor encoded by *BoPMEI1* showed expression specifically in mature pollen grains and pollen tubes. Ectopic expression of *BoPMEI1* in the antisense orientation in Arabidopsis suppressed the orthologous gene *At1g10770*, resulting in retarded pollen tube growth and sterile buds. These sterile flowers displayed stamens with shorter filaments than wild type with no pollen on the surface of the shrunken anthers [47]. Furthermore, in a study performed in *Brassica Rapa* ssp. *pekinensis* (Chinese cabbage), it was argued that the stamens in a male-sterile mutant (*ftms*) were relatively smaller than the wild-type line. The *ftms* mutant showed altered anthers without pollen. Interestingly, transcriptome analysis of the *ftms* mutant elucidated 17 differentially expressed *PMEI* genes compared to the wild type [48].

Another gene directly affecting gynoecium development is *PMEI3*; its overexpression in Arabidopsis plants causes reduced growth of the valves and thinner carpels than wild types [42].

The biochemical remodeling of pectins seems to be important. In a study performed in potato, different pectin remodeling enzymes were overexpressed, resulting in plants with pectins that contained less arabinan and rhamnogalacturonan residues and these plants showed collapsed pollen grains [49].

Furthermore, pectins can be modified by apoplastic enzymes; one is the pectinacetylesterases (PAEs) that can remove acetyl-substituents [50,51]. Their role in the development of floral tissues is scarce. However, one example is the overexpression of a pectin acetylesterase (*PAE1*) from populus in *Nicotiana tabacum* plants, which leads to a decrease of acetyl esters levels of pectin. Deacetylation produced changes in the cellular elongation of floral styles and filaments, the germination of pollen grains, and pollen tube growth [38].

Among the enzymes that degrade pectin (pectinases), the function of polygalacturonases (PGs) has been studied in several plant species and is related to anthers and pollen formation. When PG activity is absent or reduced in mutants, pollen development, pollen release, and pollen tube growth are altered [52,53,54,55].

PGs have been detected in the cytoplasm of mature pollen, pollen tubes, and the pistil of *Nicotiana benthamiana* plants. Significantly higher PG expression was present after incompatible pollination in comparison with the compatible stigma, this suggests a potential function of PGs in regulating stigma incompatibility. Furthermore, the application of exogenous PGs resulted in pollen tube growth inhibition or failure of germination [56]. Furthermore, several works focused on PGs such as the Arabidopsis orthologue of *PGDZAT/ADPG1* (*RDPG1*), *Brassica campestris Male Fertility 16* (*BcMF16*), *PGAZAT/ADPG2*, and *At1g80170* found that they are consistently expressed in male tissues such as anthers, pollen grains and dehiscence zones in the flower [57,58,59].

In another study of a putative PG called *POLYGALACTURONASE INVOLVED IN EXPANSION 1* (*PGX1*), which is very similar in its DNA sequences to the known PGs, in overexpressing Arabidopsis plants, alterations in the total activity of PGs and the composition of the cell wall were detected. The phenotype of these plants generated flowers with extra petals, which suggests the participation of *PGX1* in the early events of flower organ formation [60].

Biochemical modifications in pectins appear to be decisive in the development of reproductive structures. As it was reviewed in this section, several works have been done with regard to androecium and specifically, to the development and function of pollen grains, changes in the expression of genes that encode for pectin remodeling enzymes, and its biochemical activity is important, because it can reduce the reproductive capacity of plants. 

## 5. Other Cell Wall Proteins

Cell wall proteins play essential roles in all physiological processes and particularly in the dynamics of the components within the cell wall [61]. The WallProtBD, a database for plant cell wall proteomics, holds 188 proteins identified in inflorescence tissue from Arabidopsis [62]. Of the total proteins detected in inflorescences, a subclassification was made according to their abundance in the proteomics data. The most abundant are as follows: proteins acting on carbohydrates (25.7% of the total cell wall proteins): the most abundant in this class are Glycoside hydrolases (GHs), acting on cellulose, hemicellulose, and some pectins as mentioned in the sections before; other proteins in this class are oxido-reductases and peroxidases. Then, the following categories are as follows: proteases (11.2%), including Ser proteases, Asp proteases; proteins with interaction domains (11%), such as enzyme inhibitors and LRRs; proteins involved in signaling (6.6%), including AGPs, receptors, and structural proteins; proteins related to lipid metabolism (5.8%), i.e., lipases GDSL and lipid transferase proteins. Miscellaneous proteins (11%), such as PAPs; and finally, proteins with unknown function (12.5%) [61]. Functional information about most of these proteins is scarce, but several works have begun to unmask the participation of these proteins during reproductive development.

These proteins are located in the cell walls, but do not have a direct action on carbohydrates; functional characterization indicates a function on processes affecting floral organ formation.

An interesting example is *INFLORESCENCE DEFICIENT IN ABSCISSION* (*IDA*), which is a gene that encodes a short protein with an N-terminal secretory signal peptide. It has been detected in the extracellular space, and its signaling is carried out through a protein similar to a leucine-rich repeat-receptor-like kinase (LRR-RLK), encoded by the functionally redundant genes *HAESA* (*HAE*) and *HAESA-LIKE 2* (*HSL2*). Overexpressing IDA exhibits early abscission of floral organs [63]. This process could be related to a decrease in the expression of the cell wall remodeling enzymes, thus suggesting it is involved in cell separation during flower dehiscence, in a regulation mechanism mediated by the IDA signaling [63]. Recent work in *Nicotiana benthamiana* showed that the overexpression of *NbenIDA1A* gene generated shorter corollas, premature senescence of this whorl, and corolla abscission was also accelerated [64]. On the other hand, the double mutant *hae hsl2* generates flowers that do not open after pollination [65]. 

Within the cell wall, the interaction between ligands and receptors is mediated. A gene encoding a cell wall-associated kinase (*WAK4*) from Arabidopsis is associated with cell elongation. An antisense line of *WAK4* in Arabidopsis exhibited smaller floral buds than wild type [66]. 

The participation of miscellaneous proteins, which do not have a clear predicted molecular function, i.e., domain of unknown function (DUF) proteins, have been reported related to flower development. In Arabidopsis, four genes encoding for the DUF642 domain have been detected. *BIIDXI* (*BDX*) is the only DUF642 gene detected in the anthers, and when mutated, pollen grain viability is reduced compared to wild type [67]. Curiously, decreased pectin methylesterase activity was found in *bdx* mutants. Although the mechanism by which BDX modifies the activity of PMEs remains unknown, this phenomenon may be relevant in modifying the cell wall and thus in the formation of reproductive structures [68].

As we have seen, cell wall modifications are a regular feature of flower development. Proteins that act on the main components of the cell wall seem to be relevant, however, there are other proteins with unknown catalytic capacity that appear to alter the architecture of the cell wall through different mechanisms.

Expansins are proteins that disrupt non-covalent binding between cellulose microfibrils and other hemicelluloses. Their participation has been described mainly in the expansion of cell walls in response to increased turgor pressure within the cell [69]. Expansins have been studied in several species. *AtEXPA4* and *AtEXPB5* displayed consistent expression patterns in mature pollen grains and pollen tubes and the *atexpa4 atexpb5* double mutant was defective in pollen tube growth [70]. In petunia, the expansin encoding gene *PhEXP1* is preferentially expressed in petal limbs during development. Silencing of *PgEXP1* in petunia revealed a decrease in petal limb size, reduced epidermal cell area, and alterations in cell wall composition and structure [71]. 

The presence of expansin genes was detected in pea petals [72] and tomato flowers [73], suggesting possible participation in petal size in other flowering plants as well. In tomato, during floral abscission, a gradual increase in the detection of an anti-expansin antibody in the flower pedicel was observed, suggesting a role for expansins in the flower abscission process [28]. In carnation, *DcEXPA1* and *DcEXPA2* transcripts accumulate in petals of opening flowers. Similarly, the expansins *EgEXPA1, EgEXPA2*, and *EgEXPA3* found in *Eustoma grandiflorum* could be participating in the continuous growth of the petals during the development of the flower [74]. In *Mirabilis jalapa*, a set of expansin transcripts was identified that include α-expansins (*MjExp1-MjExp7*) and β-expansins (*MjExpB1-MjExpB3*) with detectable changes in transcript abundance during flower development, particularly in the rapid expansion and senescence of the ephemeral flowers [75]. The present findings suggest that the two expansin genes (*DcEXPA1* and *DcEXPA2*) can be associated with petal growth and development during carnation flower opening [29]. In summary, all these findings suggest the importance of expansins in flower development even when there are few functional studies in this regard.

Arabinogalactan proteins (AGPs) are extensively glycosylated hydroxyproline-rich glycol proteins found along plants. They are thought to have important functions in plant growth and development, especially in plant reproduction [76,77,78]. Several works support their participation such as the gene *LeAGP-1* in *Lycopersicum esculetum*, where overexpressing plants presented a greater number of inflorescences than wild type and most floral buds do not develop completely [79]. In Arabidopsis, *AtAGP19* showed relevance in the development of flowers and fertility. In the null mutant, besides less and smaller flowers, stamen and ovule development were affected, resulting in less seed [80]. In *Brassica campestris*, the putative *MALE FERTILITY 8* (*BcMF8*) gene encodes an AGP, which is expressed in pollen and pollen tubes. Antisense plants presented morphological defects in pollen grains and pollen tube growth, causing a lower seed yield [81].

Finally, there are studies on other proteins that seem to be good study potentials. In rice, plants with a mutation in the *RICE MALE STERILE2* (*RMS2*) gene, which encodes a GDSL esterase/lipase, showed shrunken anthers with abnormal pollen, resulting in male sterility in this *rms2* mutant. This family of proteins is related to lipid metabolism [82], however, this type of activity in the apoplast is relevant but poorly studied. 

Based on the available information, probably many of the cell wall proteins participate in flower development, making them interesting candidates to study in greater depth.

## 6. Transcriptional Control and Hormonal Modulation

A lot of enzyme families are potentially involved in the biosynthesis and modifications of cell wall components see CAZy database [83]. Within each family, the expression pattern of each member can be highly specific, as observed in studies related for instance to the shoot apical meristem (SAM) [84,85] and the root apical meristem (RAM) [86], where the expression of cell wall-related genes can be determined at good spatial resolution [11]. For floral tissues, the expression of various genes encoding cell wall-related enzymes is restricted to specific organs or tissues within the organs, suggesting a very tight control at the transcriptional level, for instance, some members of the XTH family (Figure 3). 

Most of the genes mentioned in this review encode enzymes, the last acting players in regulatory cascades. An interesting question is how their expression is regulated. Transcription factors (TFs) involved in flower development are good candidates since they often function as master regulators, causing severe developmental defects related to cell wall processes. For instance, the cell wall composition has been studied during gynoecium medial domain development [7], a zone where many changes occur to allow carpel fusion, ECM formation, and cell-death programs to form the transmitting tract [87,88]. In this region, two TFs, SEEDSTICK (STK) and NO TRANSMITTING TRACT (NTT) were identified as regulating cell wall degradation processes [89]. Other key players regulating transmitting tract formation include several bHLH TFs (SPATULA, HECATE, and HALF FILLED) [90,91]. Additionally, the master regulators control specific aspects of developmental processes related to the cell wall, such as STK in the control of the seed coat [92,93] and lignification in the funiculus for seed abscission [94]. Another example is ETTIN (ETT) as a regulator of pectin methylesterase activity in the gynoecium valves [42]. Two TFs important for flower development are AINTEGUMENTA and AINTEGUMENTA-LIKE6, and they regulate floral organ initiation and growth through modifications to the cell wall polysaccharide pectin. In the double mutant reduced levels of demethylesterified homogalacturonan were observed [95]. As a final example, two TFs important for floral meristem termination are CRABS CLAW and KNUCKLES. In the double mutant for these two genes, a reduced level of (1-4)-β-d-galactan has been reported that could be restored with introducing the auxin biosynthesis gene *YUCCA4* [96]. 

Similarly, little is known about how endogenous regulators such as hormones, including biosynthesis, transport, and perception, may actively participate in this process by controlling the expression of genes that subsequently modify cell wall components in the flower. 

Auxins are involved in cell wall regulation [97]. Cellulose distribution in the extracellular matrix contributes to establishing PIN polarity [98]. Furthermore, disruption of auxin transport results in a naked stem with no or very few lateral organs (*pin* mutant), which can be partially recovered by affecting xyloglucan biosynthesis, suggesting that this xyloglucan synthesis is partially responsible for auxin transport [10]. In other organs, such as the hypocotyl, auxin responsive gene induction and auxin synthesis induce *XTH* gene expression [99]. The importance of demethyl esterification of pectins in the shoot apical meristem is known to give rise to organ primordia through the punctual distribution of auxins [100]. Overexpressing of *PMEI3* resulted in delocalization of the PIN1 auxin transporter, and these plants had a more rigid cell wall due to lower demethyl esterification of pectins [100].

Another hormone necessary for plant development are cytokinins. Cytokinin signaling is clearly present in the carpel margin meristem (CMM) and the valve margins [101,102]. Some work has shown links between cytokinin signaling and cell wall synthesis in flowers. For example, mutations in the histidine-containing phosphotransfer factor 4 of Arabidopsis (*AHP4*) affect anther dehiscence by affecting the expression of genes related to cellulose and hemicellulose synthesis [103]. In roots, cytokinins induce genes encoding for expansins [104,105].

Although the involvement of these hormones in flower development is widely known, future work is needed to understand the relationship between the response to these and other hormones and cell wall development that contribute to flower development.

## 7. Concluding Remarks

Modifications in the cell wall are decisive in floral organ development. Alterations in cellulose synthesis, which is the most representative component in the cell wall, show floral organs that do not reach their optimal sizes for processes such as fertilization. Furthermore, loss in biomass is a relatively predictable event since cellulose is the main cell wall component. However, there are many genes involved in the cellulose synthase complex that are not yet properly identified and studied. 

Hemicelluloses modulate senescence and abscission processes. The biochemical modifications and degradation may have a participation in the development of organs. Specific expression patterns of genes encoding enzymes that synthesize/modify, or degrade hemicelluloses give indications of their relevance in specific regions of the flower and the necessity of tight transcriptional control.

Pectins are found in various events, from the formation of pollen grains, the development of fundamental structures in the gynoecium, and participation in the development of the perianth organs. The characterization of regulators that modify and degrade pectins are very representative. Alterations in the methyl esterification of pectins can modify floral structures and hence the fertilization process.

Of the proportion of proteins present within the cell wall, those that do not have enzymatic activity also seem to be very relevant for flower development. Expansins have been described in several flowering plant species. However, how they participate during development and what their importance is in collaboration with other interactors within the cell wall still needs more research. 

The proteins involved in signal transduction are very important in the life cycle of plants. Both receptors and ligands are found in the apoplast zone. Their regulation not only occurs outside the membrane, but it is also believed that genes related to modifying the biochemistry of the cell wall could also be regulated. Another type of protein with unknown function i.e., some domain of unknown function proteins (DUFs), seems to be relevant and quite visible phenotypes in loss-of-function mutants have been observed. However, it is unknown how their mechanism of action could be, it is known that they can influence enzymatic activity at least indirectly of other proteins. The expansion of the cell wall allows the cells to acquire their final size, shape, and identity in the different organs of the plant. Therefore, the synthesis and modification of the main components of the cell wall are important for the formation of the reproductive structures [8,9].

In Figure 3, it is observed how enzymes involved in the synthesis or modifications of cellulose, hemicelluloses, or pectin participate in the development of specific floral organs, showing that flower formation requires regulation of several developmental processes with significant participation of the cell wall. Comprehensive reviews have been made, indicating the potential that the cell wall has within plant development [6], including specifically in flower development [106,107], pollen tube growth [108], and fertilization [91,109]. However, the function of proteins involved in the formation, development, or arrangement of cell walls is poorly understood in floral tissues. Although, based on results of promoter activity, gene expression, or immunolocalization of cell wall components, it gives us an idea that some of them could be involved in floral organ development [6,108,110,111,112,113,114,115].

It is important to highlight that several genes that could actively participate in flower development have only been detected at the transcriptomic level without being functionally studied. On the other hand, at the proteomic level, the mechanism of action of a large proportion of proteins detected in the cell wall is still unknown, which could be relevant in scaffolding-type processes and signal transduction related to flower development.

## Figures and Tables

**Figure 1 genes-12-00978-f001:**
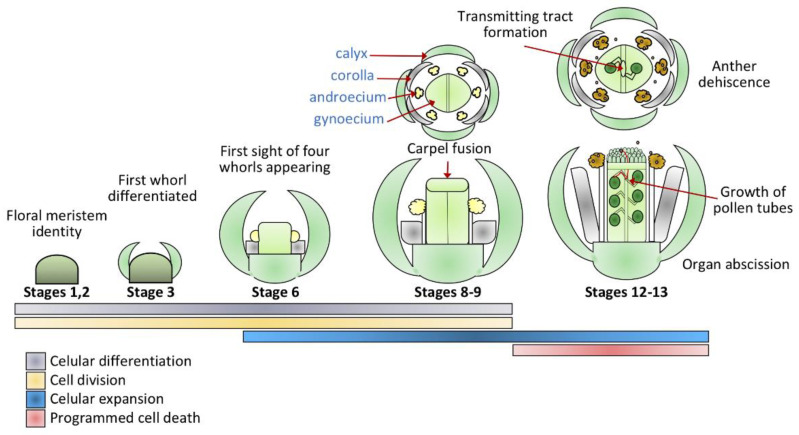
Flower development is related to cellular processes where the cell wall actively participates. In the earliest stages of flower development, cell proliferation is highly active. Up to Stage 3 of floral development, it can be seen how the calyx is already shown in the shape of a primordium, differentiated from the central structure that will give rise to the other three flower whorls. Cell proliferation and differentiation processes are essential in these steps, where the cell wall is actively involved. In Stage 6, the structures of the already differentiated floral whorls are conspicuously observed, where the sepals already cover the innermost structures. Later in Stage 8, the marginal meristem of the carpel is a structure that is already present and will give rise to tissues such as the placenta and ovules. For Stage 9, a shaped septum can be perfectly recognized. At Stage 12, the style and transmitting tract are differentiated as well as the valves, and the margins of the valves begin to be morphologically distinct. The petals and the stamens are structures where the cellular expansion is determinant for the growth of the organs. In Stage 13, the gynoecium is fully developed, with anthesis and self-pollination of the flower taking place. Programmed cell death is detected in the abscission zones where the organs open first by the separation of the cells.

**Figure 2 genes-12-00978-f002:**
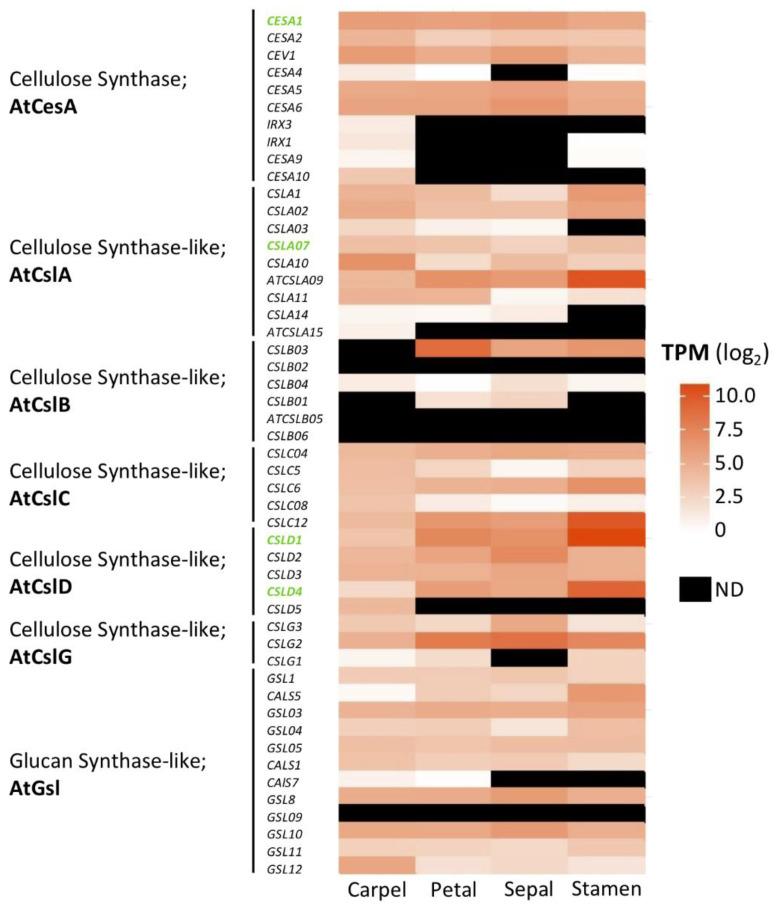
Expression of cell wall biosynthesis genes in Arabidopsis floral tissues. List of genes belonging to the cell wall biosynthesis family (TAIR database). Orange scale color indicates gene expression levels in floral tissues; white color indicates no expression, and black no data (ND). Values indicate log2-transformed transcripts per million (TPM) for each tissue; data from [23]. Gene names in green color mark genes that are functionally characterized.

**Figure 3 genes-12-00978-f003:**
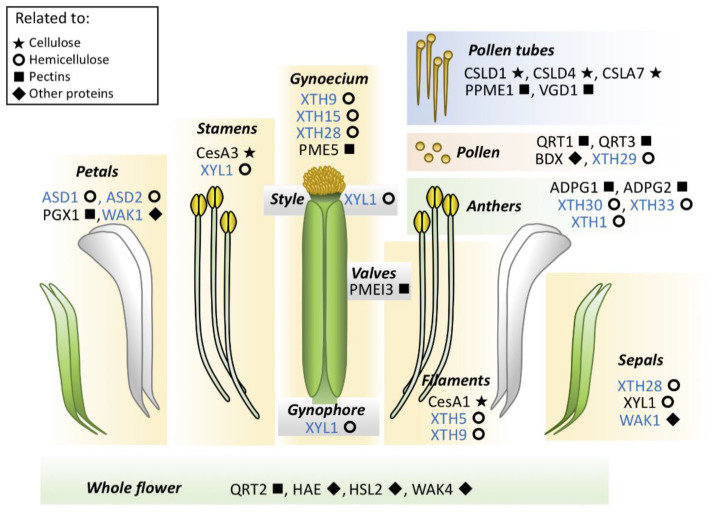
Cell wall-related proteins with a function in the Arabidopsis flower. Functional studies revealed the involvement of enzymes related to cellulose, hemicelluloses, or pectin biosynthesis or modifications during flower development. The function of XTH family members (names in blue color) has only been characterized at the gene expression level. Symbols besides the enzyme name indicate relation with cellulose (stars), hemicelluloses (circles), pectins (squares); or other proteins non-acting on carbohydrates (diamonds). ASD1,2, α-L-ARABINOFURANOSIDASE1, 2; PGX1; POLYGALACTURONASE INVOLVED IN EXPANSION1; WAK1, WALL-ASSOCIATED KINASE1; CesA3, CELLULOSE SYNTHASE A3; CesA1, CELLULOSE SYNTHASE A1; XYL1, XYLOSIDASE 1; PME5, PECTIN METHYL ESTERASE 5; PMEI3, PECTIN METHYL ESTERASE INHIBITOR 3; CSLD1, 4, CELLULOSE SYNTHASE-LIKE D1, 4; CSLA7, CELLULOSE SYNTHASE-LIKE A7; PPME1, PECTIN METHYL ESTERASE 1; VGD1, VANGUARD1; QRT1,2,3, QUARTET1,2,3; BDX, BIIDXI; ADGP1,2, ARABIDOPSIS DEHISCENCE ZONE POLYGALACTURONASE1,2; HAE, HAESA; HSL2, HAESA-LIKE 2.

## Data Availability

Not applicable.

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
