# Peer review of "Building a Flower: The Influence of Cell Wall Composition on Flower Development and Reproduction"

_genes, 2021, doi:10.3390/genes12070978_

Round 1

Reviewer 1 Report

The review presents an extensive and timely overview of what is known about genes that regulate cell wall synthesis and their impact on floral organ development. It is a very interesting read that links plant development with cell wall research. I think the review is well structured with a defined division into the major cell wall components. The figures are clear and help to summarize and understand the existing literature.

I have few suggestions and comments.

The paragraph in line 77 to 83  looks a bit disjoint from the rest. I feel it could be improved removing ''Using the triple mutant of the genes'' and starting directly describing the genes function in the wall and  explaining straight away why their phenotypes are interesting on a developmental point of view.

In line 111 and 112 I feel the authors could add that a lack of cellulose synthesized can impact cell growth and results in smaller organs instead of referring to it only as biomass.

In line 248 The authors need to remove the word affectation and leave only visible effects.

I think the concluding remarks could be improved by adding few lines to summarize the importance of cell wall for cell growth and shape and how it ultimately influence cell organ shape and size.

Author Response

R: Thank you for time and nice words.

  • The paragraph in line 77 to 83 looks a bit disjoint from the rest. I feel it could be improved removing ''Using the triple mutant of the genes'' and starting directly describing the genes function in the wall and explaining straight away why their phenotypes are interesting on a developmental point of view.

R: The changes were done according to the suggestions of the author.

  • In line 111 and 112 I feel the authors could add that a lack of cellulose synthesized can impact cell growth and results in smaller organs instead of referring to it only as biomass.

R: The changes were done according to the suggestions of the author. Now it reads: ´These phenotypic alterations suggest an impact in cell growth with a lower amount of cellulose synthesized and a lower amount of generated biomass, resulting in smaller calyx and corolla whorls.´

  • In line 248 The authors need to remove the word affectation and leave only visible effects.

R: Done

  • I think the concluding remarks could be improved by adding few lines to summarize the importance of cell wall for cell growth and shape and how it ultimately influences cell organ shape and size.

R: Thank you. We have added the lines: ´ The expansion of the cell wall allows the cells to acquire their final size, shape, and identity in the different organs of the plant. Therefore, the synthesis and modification of the main components of the cell wall are important for the formation of the reproductive structures [8,9].´

Reviewer 2 Report

The present review aims to highlight the importance of the cell wall in the flower development. In particular some of the cell wall-synthesizing and -modifying enzyme families are well described and their putative functions during flower development are discussed. In details, the three main polysaccharides of the cell wall (cellulose, hemicelluloses and pectins) and the proteins regulating their structure are described with a focus in the flower development. Additionally, other cell wall proteins are presented for their putative function in cell wall modification and flower development, as well as the main transcriptional and hormonal regulators which can control the cell wall metabolism during flower development. 

Overall this review is well structured with the advantage to present together the three main polysaccharides of the primary cell wall in the context of flower development. However the title should be changed. Indeed, the functions of the enzymes acting on cell wall polymers are presented with examples in several reproductive events instead of only flower development. In my sense, this review is more related to flower development and reproductive events that result. The abstract should be also changed to be in accordance with something more general in the flower development as well as reproduction.

In detail, I have some points that need attention:

  • Line 61: “polymers” word is more appropriated than “molecules” in the sentence
  • Figure 2: In the legend, each abbreviation should be defined, in particular for gene families not mentioned in the main text as well as TPM which is not defined. It can be nice to add color code for the name of the genes, in order to distinguish families on the heatmap. For instance, one color can be used for CSL and other one for GSL. It could be also interesting to make the same kind of heatmap, if the data are available, for the genes encoding enzymes involved in the metabolism of hemicelluloses and pectins with the same focus on flower development.
  • Line 151: “sugars” word is more appropriated than “saccharides” in the sentence
  • Line 223-225: the backbone of HGs and RGs is composed of “galacturonic acids”. “galatose” has to be replaced by galacturonic acid.
  • Lines 228-229: The sentence should be changed, because it is not really clear. With the current sentence we can believe that methylesterification occurs after synthesis/secretion into the cell wall whereas every steps of the synthesis occur in the Golgi apparatus. Another sentence could be “HGs consist to linear backbones of Gal-A that are firstly polymerized by Golgi-located enzymes such as GAUTs and GATLs, then highly methylesterified at their C-6 carboxyl group by other Golgi-located enzymes with the pectin methyltransferases (PMT)”.
  • Line 234: “the complete” should be removed because we do not know if the degradation is total.
  • Lines 285-290: deacetylation by PAE is mentioned in this paragraph but acetylation of pectins is not mentioned before in the part related to pectins. A short paragraph should be added when pectin synthesis and remodeling are described
  • Lines 292-301: In this paragraph, the role of some PGs is discussed. Unfortunately, some interesting papers about PGs and abscission are not developed. It could be nice to add some details about PGAZAT/ADPG2 for instance.
  • In the fifth part about other cell wall proteins, the development about structural protein such as AGPs and extensins is missing. There are some papers about AGPs and reproduction for instance. It could be interesting to add some details about these glycoproteins, which have important function in the regulation of the cell wall.
  • Line 355: “unknown” word could be more appropriated than “not known” in the sentence
  • Figure 3: lines 404-405 of the legend, there are some problem with the symbols that are not visible between the brackets.

Author Response

Overall this review is well structured with the advantage to present together the three main polysaccharides of the primary cell wall in the context of flower development. However, the title should be changed. Indeed, the functions of the enzymes acting on cell wall polymers are presented with examples in several reproductive events instead of only flower development. In my sense, this review is more related to flower development and reproductive events that result. The abstract should be also changed to be in accordance with something more general in the flower development as well as reproduction.

R: Thank you for your time and nice words. As suggested by the reviewer, we changed the title. Now it reads: ´Building a flower: The influence of cell wall composition on flower development and reproductive features.´

In detail, I have some points that need attention:

  • Line 61: “polymers” word is more appropriated than “molecules” in the sentence

R: Done

  • Figure 2: In the legend, each abbreviation should be defined, in particular for gene families not mentioned in the main text as well as TPM which is not defined. It can be nice to add color code for the name of the genes, in order to distinguish families on the heatmap. For instance, one color can be used for CSL and other one for GSL. It could be also interesting to make the same kind of heatmap, if the data are available, for the genes encoding enzymes involved in the metabolism of hemicelluloses and pectins with the same focus on flower development.

R: Figure legend was modified; TPM explained. Some labels were added to the figure to classify enzymes into sub-families, as suggested.

  • Line 151: “sugars” word is more appropriated than “saccharides” in the sentence

R: Done

  • Line 223-225: the backbone of HGs and RGs is composed of “galacturonic acids”. “galatose” has to be replaced by galacturonic acid.

R: Done

  • Lines 228-229: The sentence should be changed, because it is not really clear. With the current sentence we can believe that methylesterification occurs after synthesis/secretion into the cell wall whereas every steps of the synthesis occur in the Golgi apparatus. Another sentence could be “HGs consist to linear backbones of Gal-A that are firstly polymerized by Golgi-located enzymes such as GAUTs and GATLs, then highly methylesterified at their C-6 carboxyl group by other Golgi-located enzymes with the pectin methyltransferases (PMT)”.

R: The changes were done according to the suggestions of the author. Now it reads: ´Pectins are synthesized in the Golgi apparatus to be methyl esterified after being methyl esterified synthesized at their C-6 carboxyl group by pectin methyltransferases (PMT).´

  • Line 234: “the complete” should be removed because we do not know if the degradation is total.

R: Done.

  • Lines 285-290: deacetylation by PAE is mentioned in this paragraph but acetylation of pectins is not mentioned before in the part related to pectins. A short paragraph should be added when pectin synthesis and remodeling are described

R: The addition has been made as suggested. The following lines have been added: ´Another chemical change that pectins can undergo are the O-acetylatation at the C-2 or C-3 position. The resulting acetylesters change dynamically during the growth and development of plants [Gou et al.].´

  • Lines 292-301: In this paragraph, the role of some PGs is discussed. Unfortunately, some interesting papers about PGs and abscission are not developed. It could be nice to add some details about PGAZAT/ADPG2 for instance.

R: Additions to the texts have been made, including references, as suggested by the reviewer. New text: ´PGs have been detected in the cytoplasm of mature pollen, pollen tubes, and the pistil of Nicotiana benthamiana plants. Significantly higher PG expression was present after in-compatible pollination in comparison with the compatible stigma, this suggests a potential function of PGs in regulating stigma incompatibility. Furthermore, the application of exogenous PGs resulted in pollen tube growth inhibition or failure of germination [Liao et al 2020]. Furthermore, several works focused on PGs such as the Arabidopsis orthologue of ADPG1 (RDPG1), Brassica campestris Male Fertility 16 (BcMF16), PGAZAT, PGDZAT, and At1g80170 found that they are consistently expressed in male tissues such as anthers, pollen grains and dehiscence zones in the flower [Sander et al., 2001; González-Carranza et al., 2007; Zhang et al., 2012].´

  • In the fifth part about other cell wall proteins, the development about structural protein such as AGPs and extensins is missing. There are some papers about AGPs and reproduction for instance. It could be interesting to add some details about these glycoproteins, which have important function in the regulation of the cell wall.

R: Additions to the texts have been made, including references, as suggested by the reviewer. New text: ´Arabinogalactan proteins (AGPs) are extensively glycosylated hydroxyproline-rich glycol proteins found along plants. They are thought to have important functions in plant growth and development, especially in plant reproduction [Showalter, 2001; Coimbra et al, 2007; Silva et al, 2020]. Several works support their participation such as the gene LeAGP-1 in Lycopersicum esculetum, where overexpressing plants presented a greater number of inflorescences than wild type and most floral buds do not develop completely [Sun et al., 2004]. In Arabidopsis, AtAGP19 showed relevance in the development of flowers and fertility. In the null mutant, besides less and smaller flowers, stamen and ovule development was affected, resulting in less seed [Yang et al., 2007]. In Brassica campestris, the putative MALE FERTILITY 8 (BcMF8) gene encodes an AGP, which is expressed in pollen and pollen tubes. Antisense plants presented morphological defects in pollen grains and pollen tube growth, causing a lower seed yield [Lin et al., 2014]. ´

  • Line 355: “unknown” word could be more appropriated than “not known” in the sentence.

R: Done

  • Figure 3: lines 404-405 of the legend, there are some problem with the symbols that are not visible between the brackets.

R: Figure legend was modified to solve this issue.